# High-Density Lipoproteins at the Interface between the NLRP3 Inflammasome and Myocardial Infarction

**DOI:** 10.3390/ijms25021290

**Published:** 2024-01-20

**Authors:** Helison R. P. Carmo, Isabella Bonilha, Joaquim Barreto, Massimiliano Tognolini, Ilaria Zanotti, Andrei C. Sposito

**Affiliations:** 1Atherosclerosis and Vascular Biology Laboratory (Aterolab), Division of Cardiology, State University of Campinas (UNICAMP), Campinas 13084-971, SP, Brazil; helison.rafael@gmail.com (H.R.P.C.); isaoliveira.ib@gmail.com (I.B.); joaquimbarretoantunes@gmail.com (J.B.); sposito@unicamp.br (A.C.S.); 2Department of Food and Drug, University of Parma, 43124 Parma, Italy; massimiliano.tognolini@unipr.it

**Keywords:** HDL, NLRP3 inflammasome, myocardial infarction, ischemia-reperfusion injury

## Abstract

Despite significant therapeutic advancements, morbidity and mortality following myocardial infarction (MI) remain unacceptably high. This clinical challenge is primarily attributed to two significant factors: delayed reperfusion and the myocardial injury resulting from coronary reperfusion. Following reperfusion, there is a rapid intracellular pH shift, disruption of ionic balance, heightened oxidative stress, increased activity of proteolytic enzymes, initiation of inflammatory responses, and activation of several cell death pathways, encompassing apoptosis, necroptosis, and pyroptosis. The inflammatory cell death or pyroptosis encompasses the activation of the intracellular multiprotein complex known as the NLRP3 inflammasome. High-density lipoproteins (HDL) are endogenous particles whose components can either promote or mitigate the activation of the NLRP3 inflammasome. In this comprehensive review, we explore the role of inflammasome activation in the context of MI and provide a detailed analysis of how HDL can modulate this process.

## 1. Introduction

Owing to its unique role, the myocardium perpetually engages in electromechanical and biochemical metabolic activities, leading to a significant oxygen consumption per unit of tissue [1]. Consequently, the myocardium encounters substantial hurdles when confronted with ischemia, a condition characterized by limited oxygen supply, with its severity influenced by various factors, including the duration and extent of the at-risk myocardial tissue [2]. Furthermore, the presence of cardiometabolic disorders can alter the accessibility of energy substrates, including fatty acids, glucose, and ketones, thereby exacerbating the ischemic injury [3].

The ultimate consequence of prolonged cardiac ischemia is myocardial infarction (MI), a condition accountable for one-third of global fatalities. The extent of MI is a potent predictor of various adverse cardiovascular events, encompassing mortality, recurrent MI, arrhythmias, congestive heart failure, angina, and the need for revascularization, as substantiated by several studies [4,5,6,7]. Consequently, there has been widespread adoption of abbreviated reperfusion therapies aimed at attenuating the extent of MI. In other words, reperfusion therapies aim to restore blood flow by unclogging an occlusive thrombus, mostly resulting from the rupture of an atherosclerotic plaque. This is achieved through two primary methods: non-invasive procedures, which involve the use of thrombolytic drugs, and invasive procedures, such as primary percutaneous coronary intervention (PCI) or coronary artery bypass graft (CABG) surgery. These methods allow for blood reperfusion and aid in the restoration of appropriate circulation. Notably, all existing research consistently demonstrates an inverse relationship between the duration of reperfusion therapy and both short-term and long-term mortality rates, underscoring the significance of prompt intervention [8].

Consequently, early and effective reperfusion therapies have become a cornerstone in reducing mortality and morbidity associated with MI. Yet, reperfusion itself triggers a cascade of molecular mechanisms that contribute to additional myocardial damage, collectively known as myocardial ischemic and reperfusion injury (IRI) [9,10]. From the mechanistic point of view, following reperfusion, there is a rapid intracellular pH shift, disruption of ionic balance, heightened oxidative stress, increased activity of proteolytic enzymes, initiation of inflammatory responses, and activation of several cell death pathways, encompassing apoptosis, necroptosis, and pyroptosis [11]. These metabolic changes activate the intracellular sensor responsible for mediating the local immune system response. This sensor is formed by nucleotide oligomerization domain (NOD)-like receptors, leucine-rich repeat (LRR)-containing receptors (NLRs), and the multiprotein signaling complex named the NLRP3 inflammasome, and is made of pyrin domain-containing 3 (NLRP3), adaptor apoptosis-associated speck-like proteins containing a CARD (ASC), and pro-caspase-1 [12]. The NLRP3 inflammasome is an intracellular defender that acts against the cellular fragments and cytoplasmic contents of injured and/or dead cells, called damage-associated molecular patterns (DAMPs) (also known as alarmins).

Indeed, MI activates the NLRP3 inflammasome through the DAMPs tissue dissemination, resulting in two deleterious mechanisms: (i) the activation of proinflammatory interleukins (pro-IL-1β and pro-IL18), (ii) and lytic cell death due to the pore formation on the cellular membrane, characterized by proinflammatory cell death or pyroptosis. Regarding the secretion of proinflammatory cytokines, the activation of the NLRP3 inflammasome leads to pathogenicity in various resident cardiac cells. The increased activity of the NLRP3 inflammasome leads to an exacerbated production of IL1β and IL18, responsible for an acute immune and inflammatory response. This effect can manifest locally as myocardial inflammation or systemically targeting the vascular endothelium and multiple organs. The presence of IL1 receptors in immune cells facilitates the paracrine and autocrine effects of these proteins, potentially inducing the increased transcription of proinflammatory genes, amplifying their synthesis and secretion [10]. Detrimental clinical effects such as fever, hypotension, myocardial contractility dysfunction, and increased ischemic events are mainly associated with elevated plasma IL1β levels. However, insights from in vitro experiments indicate that IL-1β secretion is primarily associated with cardiac fibroblasts, whereas IL-18 secretion is more closely linked to cardiomyocytes [13,14]. Other proinflammatory plasmatic markers such as high-mobility group protein B1 (HMGB1), leukotrienes, prostaglandins, chemokines, interferon gamma (IFN-γ), necrosis factor-alpha (TNF-α), interleukin 6 (IL-6), cyclooxygenase-2, intercellular adhesion molecule-1 (ICAM-1), and vascular cell adhesion molecule-1 (VCAM-1) have also been associated with pyroptosis damage at the systemic level [12,15,16]. A comprehensive overview of the suggested cellular mechanisms involved in these interactions is provided in Figure 1.

Studies conducted in animal models of MI estimate that IRI contributes to as much as 50% of the final infarct size (Figure 2) [9]. These ex vivo studies also shed light on diverse strategies with the potential to alleviate IRI, either by the inhibition of apoptosis or pyroptosis pathways [17,18]. However, the translation of their findings into clinical benefits has proven to be challenging. This difficulty hints at a complex interplay between IRI and both the multitude and the redundancy of systems at play in the MI scenario. In this regard, both our team and other researchers have reported that coronary reperfusion with high-density lipoprotein (HDL) obtained from healthy volunteers can reduce the extent of MI [19,20,21]. Conversely, we have observed that reperfusion with HDL obtained from MI patients not only fails to provide protection but exacerbates the extent of the MI [22]. In line with this finding, we found a direct association between plasma HDL concentration and the estimated size of MI in patients as assessed by troponin curves and cardiac magnetic resonance imaging [22]. Collectively, these findings underscore the significance of HDL-related molecules in influencing cell death pathways, both positively and negatively, and emphasize their relevance within the pathophysiological construct of IRI.

In this review, we examine the relevance of innate immune signaling driven by the NLRP3 inflammasome in shaping the extent of MI and provide an in-depth analysis of the complex dual interplay involving HDL.

## 2. The Mechanisms Underlying the MI-Induced Activation of NLRP3 Inflammasome Signaling

The detection of DAMPs is a fundamental aspect of the innate immune system’s ancient defense mechanism against invaders. It not only counters threats but also notifies neighboring cells through proinflammatory cytokines. In MI patients, the initiation of proinflammatory signaling through DAMPs may occur as a consequence of IRI [10,23]. Additionally, the activation of this signaling could potentially pre-exist and be driven by modified lipoproteins like oxidized low-density lipoproteins (oxLDL), which have the potential to enhance proinflammatory responses [24]. Nevertheless, the antioxidant function of the paraoxonase family, notably paraxonase-1 (PON1) within HDL, significantly mitigates the inflammatory impact mediated by ox-LDL. Animal model experiments have indicated a direct correlation between the absence of the PON1 gene and inflammatory responses. In humans, plasma levels of PON1 exhibit an inverse correlation with the development of atherosclerosis [25].

Pattern-recognition receptors (PRRs) play a pivotal role in mediating signaling within the innate immune defense system, triggering proinflammatory responses upon detecting DAMPs. Notably, PRRs are not limited to immune cells but are also found in non-immune cardiac cells, including cardiac fibroblasts, cardiomyocytes, and endothelial cells [26,27,28,29]. Proinflammatory cytokines can act in a paracrine manner via PRRs, amplifying and intensifying signals along the same inflammasome pathway [30].

These receptors can be categorized into two subgroups: (i) transmembrane receptors, such as Toll-like receptors (TLRs) and C-type lectin receptors (CLRs) [31], and (ii) intracellular sensors, including cytoplasmic nucleotide-binding oligomerization domain-like receptors (NLRs), retinoic acid-inducible gene I-like receptors (RLRs), absent in melanoma 2-like receptors (ALRs), and cytosolic DNA receptors [32]. However, among these, the TLR and NLR families are most extensively characterized in cardiac cells, with particular focus on members like TLR4 and the NLRP3 inflammasome (Figure 1).

While the NLRP3 inflammasome serves a protective role against harmful stimuli, its dysregulated activity can paradoxically intensify proinflammatory signaling and exacerbate MI damage, ultimately leading to cell death [33]. In fact, NLRP3 inflammasome-mediated inflammation is initiated in the early stages of myocardial ischemia and can persist for days, exerting a profound influence on chronic outcomes like cardiac remodeling and congestive heart failure [34,35].

In preclinical studies modeling myocardial IRI, it has been observed that just minutes of reperfusion can trigger NLRP3 inflammasome activation [36]. Interestingly, the activation kinetics of the NLRP3 inflammasome in the heart after IRI display a gradual increase over a 6 to 24 h period during reperfusion [36]. This sustained activation contributes significantly to the inflammatory response and the severity of myocardial damage [36].

The most extensively studied pathways that lead to the activation of the NLRP3 inflammasome triggered by the MI include the following: (i) the production of reactive oxygen species (ROS) derived from the mitochondria, (ii) lysosomal destabilization, (iii) ionic imbalance (mainly presented by K^+^ efflux and Ca^2+^ mobilization), and, more recently, (iv) intracellular signaling through the adenosine triphosphate (ATP)-activated P2 purinergic receptor (P2X7R) [13].

Mitochondrial ROS production arises from a complex interplay of sources, including electron leakage from complex I and II, leading to the formation of superoxide, as well as the generation of hydrogen peroxide, which can induce the creation of other reactive species [37]. This cascade of ROS generation can trigger the activation of the NLRP3 inflammasome, a process facilitated by its interaction with thioredoxin-interacting protein (TXNIP), a member of the α-arrestin superfamily of proteins found ubiquitously in non-ischemic myocardial tissue [38] (refer to Figure 1). Notably, TXNIP itself becomes activated by ROS, setting in motion a positive feedback loop that further amplifies ROS production [39]. In agreement with these findings, studies using mice with a cardiomyocyte-specific knockout of the TXNIP gene have demonstrated a reduction in the extent of MI [40].

Lysosomal destabilization represents another crucial mechanism to consider within the context of MI [41]. This process involves the release of cathepsin B hydrolase, a lysosomal cysteine protease belonging to the papain family, from the lysosome into the cytosol. This release contributes significantly to the activation of the NLRP3 inflammasome and, consequently, exacerbates myocardial damage [42].

Mitochondrial Ca^2+^ overload has emerged as a contributor to the activation of the NLRP3 inflammasome during ischemia [43]. An excessive or sustained increase in mitochondrial Ca^2+^ levels can lead to mitochondrial damage and cell death [44]. During MI, damaged cardiac cells can release ATP from their membranes, initiating the activation of the P2X7R. This receptor, an ATP-gated extracellular ion channel known for its involvement in transmembrane ion migration [45,46], responds to ATP by promoting the influx of Ca^2+^ and Na^+^ while inducing K^+^ efflux. This classic signaling pattern triggers the activation of the NLRP3 inflammasome [39] (Figure 1). Consequently, extracellular ATP binding to P2X7R rapidly disrupts the cytosolic ion concentration gradient.

The details of K^+^ efflux in NLRP3 inflammasome activation remain less comprehensively understood and appear to be independent of Ca^2+^ mobilization. For instance, Katsnelson et al. [47] disrupted the cytosolic concentration gradient by employing BAPTA, a potent Ca^2+^ chelator and cytosolic Ca^2+^ buffer. Their findings demonstrated that cellular K^+^ efflux during NLRP3 activation operates independently of intracellular Ca^2+^ dynamics [45].

Furthermore, the P2X7R has been identified as a critical mediator of NLRP3 inflammasome activation through its interaction with the serine-threonine kinase NIMA-related Kinase 7 (NEK7), a member of the mammalian NIMA-related kinases (Nek protein) family [48]. Recently, using an in vivo animal model of MI, Yanqin Li and colleagues [49] revealed that an upregulation of P2X7R, NEK7, and NLRP3 inflammasome activation occurs after MI, and is accompanied by a deterioration in cardiac function and an increase in inflammatory markers.

## 3. The Mechanistic Basis of NLRP3 Inflammasome Priming and Activation

The NLRP3 inflammasome is expressed in various cardiovascular cells, such as endothelial cells and cardiomyocytes. Among several identified inflammasomes, including NLRP1, NLRC4, and AIM2, NLRP3 is the most extensively studied in MI pathogenesis. In a 2009 study by Iyer et al. [50] using a necrotic cell model, NLRP3 inflammasome activation was demonstrated, leading to IL-1β release. Subsequently, Kawaguchi et al. [51] emphasized the significance of NLRP3 inflammasome activity in cardiac fibroblasts during IRI.

The pro-inflammatory signaling triggered by DAMPs in MI involves two phases, termed signal 1 and signal 2 [39], both dependent on the NLRP3 inflammasome. Signal 1, also known as priming, initiates when DAMPs bind to TLRs, initiating downstream signaling [43]. In mammals, 13 TLRs are recognized [52,53], but it remains uncertain whether all are present in myocardial tissue. Most of them (TLR-2 to 7 and 9) have been identified through mRNA expression in murine cardiac cells in vitro and in vivo [54]. In humans, TLR1 to 10 have been identified thus far, with recent findings indicating increased expression in the Tlr2, Tlr3, Tlr4, and Tlr9 genes in chronic ischemic diseases [55].

To date, the most comprehensive signal description involves TLR4-mediated nuclear factor kappa B (NF-κB) signaling. DAMPs activate TLR4, initiating an intracellular signaling cascade that, in turn, activates the adapter protein myeloid differentiation primary response protein (MyD88). MyD88 subsequently triggers downstream signaling by binding to interleukin-1 receptor-associated kinase 4 (IRAK4), followed by IRAK1/2, resulting in the formation of a large complex called the Myddosome [31]. This Myddosome complex associates with the ubiquitin ligase E3 TNF receptor-associated factor 6 (TRAF6), leading to the activation of the kappa light polypeptide gene enhancer nuclear factor in the B-cell inhibitor alpha (IκBα) within the IKK complex, and subsequent translocation, along with NF-κB, into the cell nucleus. NF-κB serves as the principal transcription factor driving the proinflammatory priming response during MI [56]. Consequently, priming signaling targets gene transcription for NLRP3 inflammasome proteins (e.g., ASC and pro-caspase-1) and pro-inflammatory interleukins (pro-IL1-β and pro-IL18) [57,58,59,60] (Figure 1).

Furthermore, alternative pathways for the transcriptional activation of the priming effect may occur directly through IL-1β via the IL-1 receptor (IL1R), also coupled to MyD88, leading to transcriptional signaling via NF-κB. It is plausible that multiple non-canonical pathways exist to indicate pro-inflammatory priming, potentially involving enzymatic deubiquitination. However, our current understanding of this mechanism primarily pertains to immune cells rather than myocardial cells, necessitating further investigations.

Signal 2, also referred to as activation, is a process initiated by DAMPs, leading to the assembly or oligomerization of the NLRP3 complex. This activation is driven by proximity-induced autocatalytic activity [12]. In the presence of DAMPs, NLRP3 interacts with ASC through PYD interactions and pro-caspase-1 through CARD interactions. It is worth noting that ASC serves as a crucial scaffold protein necessary for recruiting the pro-caspase-1 effector enzyme to form the NLRP3 inflammasome [61]. Consequently, the activation of the NLRP3 inflammasome leads to the mediation of caspase-1 activity, which is a cysteine protease with specificity for aspartic residues and is classified among the proinflammatory caspases, including caspases 1, 4, 5, 11, and 12 [62] (Figure 1).

The NLRP3 inflammasome exhibits a multi-target nature, attributed to its possession of three distinct domains: (i) an LRR domain, (ii) a NACHT domain, and (iii) an amino-terminal PYRIN (PYD) domain [7]. This unique feature renders NLRP3 susceptible to the stimulation of various DAMPs. Nevertheless, the precise mechanisms linking the NLRP3 inflammasome to these DAMPs remain unclear [63]. It is widely acknowledged that the primary mode of NLRP3 inflammasome activation involves the release of pro-inflammatory cytokines via the formation of membrane pores, a process termed pyroptosis [64] (Figure 1). Inhibiting NLRP3 inflammasome activity using a caspase-1-specific probe can significantly safeguard the myocardium against IRI, mitigating the harmful consequences of pyroptosis. Consequently, pyroptosis emerges as the principal outcome of NLRP3 inflammasome activation.

## 4. NLRP3 Inflammasome Activation and Pyroptosis after Coronary Reperfusion

Historically, MI treatment primarily focused on mitigating two forms of cell death. The first is necrosis, primarily stemming from traumatic events that disrupt metabolic homeostasis, activating independent cellular pathways [65,66]. The second form is apoptosis, characterized by programmed cell death driven by intrinsic or extrinsic molecular pathways. While apoptosis serves adaptive functions, its dysregulation can lead to lethal outcomes. Pyroptosis, a cell death triggered by NLRP3 inflammasome activation, emerges as an alternative cell death pathway, presenting a distinctive challenge in MI management. Coined by Cookson and Brennan in 2001, pyroptosis, derived from the Greek word’s “pyro” meaning fire and “ptosis” meaning fall or death, was initially observed in macrophages during bacterial infection processes [67,68]. Remarkably, pyroptosis exhibits characteristics of both apoptosis, such as nuclear condensation and DNA fragmentation, and necrosis, including plasma membrane rupture and an inflammatory response [69,70]. Pyroptosis underpins the systemic elevation of IL-1β. This cytokine is associated with a wide array of inflammatory conditions, including rheumatoid arthritis, osteoarthritis, chronic obstructive pulmonary disease, asthma, inflammatory bowel disease (including Crohn’s disease and ulcerative colitis), multiple sclerosis, Alzheimer’s disease, atherosclerosis, and type 2 diabetes [27,71].

Pyroptosis entails the activation of caspase-1 and the subsequent cleavage of gasdermin-D (GSDM-D), enabling the secretion of pro-inflammatory interleukins. Hence, pyroptosis is widely recognized as a caspase-1-dependent cell death program. The cleavage of GSDM-D leads to the release of its active N-terminal fragment, which relocates to the plasma membrane’s lipid region, instigating oligomerization and pore formation. These membrane pores facilitate the outward release of active interleukins and result in lytic cell death due to the leakage of cytosolic contents [72] (Figure 1).

Preclinical studies have convincingly demonstrated the molecular signaling pathway of pyroptosis mediated by NLRP3/ASC/caspase-1 in the context of myocardial damage following MI [61]. Our research has showcased that the attenuation of pyroptosis molecular signaling via the use of the highly selective caspase-1 inhibitor VX-765 can significantly diminish infarct size in an ex vivo model of isolated perfused hearts [18]. Consistently, Yang et al. [73] demonstrated that treatment with the same caspase-1 inhibitor mitigated myocardial damage in vivo. Likewise, several other inhibitors are also capable of mitigating or inhibiting the pyroptosis pathway by targeting the NLRP3 inflammasome, caspase-1, and interleukins (specifically IL-1β and IL-18). This intervention leads to myocardial protection, as confirmed by the assessment of biological markers, including proinflammatory cytokines and Troponin I [74]. These experimental protocols suggest that the NLRP3/ASC/caspase-1 molecular signaling pathway could serve as a promising therapeutic target for mitigating pyroptosis.

## 5. The Non-Canonical Activation of Pyroptosis during MI

The non-canonical mechanisms governing inflammasome activation and its connection to pyroptosis remain partially elucidated, yet they offer a potential alternative pathway to the previously described ones. Initial investigations, utilizing lipopolysaccharide (LPS) stimulation, have unveiled immune defense system activation through mechanisms under the control of other members of the inflammatory caspase family. Specifically, in humans, caspase-4 and 5 have been implicated, whereas in mice, caspase-11 is a key player, and in both species, caspase-12 contributes to this process [75,76] (Figure 1). There is a plausible notion that specific inflammatory caspases may assume an alternative function in the proteolysis of the GSDM-D protein, which was previously regarded as the exclusive role of caspase-1 [77].

Caspase-4 and 5 in humans exhibit similarities to caspase-11 in mice, thereby classifying them as orthologs [78]. Kayagaki et al. [79] demonstrated that murine macrophages, when exposed to LPS, activate caspase-11, leading to the proteolytic cleavage of GSDM-D. This constitutes the primary mechanism of pyroptosis, as GSDM-D releases an N-terminal fragment, instigating membrane pore formation, which may indirectly activate the NLRP3 inflammasome. Likewise, in an experimental model of endotoxic shock in mice, LPS stimulation directly triggers caspase-11 activation, which, in turn, facilitates the activation of IL-1β by cleaving the transmembrane protein pannexin-1 (Panx-1) and allowing its release [80]. In the context of MI, where the classic experimental protocol of coronary artery occlusion is employed to induce in vivo myocardial IRI, an observed elevation in caspase-4 activity and GSDM-D expression underscores the potential pathophysiological relevance of this non-canonical mechanism for myocardial damage [81].

Several studies have indicated the potential role of endoplasmic reticulum (ER) stress as a central hub for alternative caspase activation [82,83]. One plausible mechanism involves the signaling of the inositol 1,4,5-triphosphate receptor (IP3R1) within the endoplasmic reticulum (ER), a crucial regulator of Ca^2+^ transport. This signaling pathway has been implicated in modulating the NLRP3 inflammasome pathway during myocardial injury [84]. Excessive Ca^2+^ accumulation, functioning as a DAMP, may potentially serve as the link mediating this connection and triggering pyroptosis.

Moreover, it is conceivable that stress triggered by diverse DAMPs might activate alternative pathways that culminate in pyroptosis and cytokine release, bypassing the primary involvement of NLRP3 inflammasome activity [84]. Nevertheless, compelling evidence indicates that in instances of prolonged pathological events, the continued presence of DAMPs is likely to ultimately lead to the canonical activation of the NLRP3 inflammasome [85].

## 6. The Double-Edged Effects of HDL Particles in MI

The plasma concentration of HDL particles, measured by either their cholesterol content or apolipoprotein A1 levels, is a widely recognized robust predictor of cardiovascular events. This phenomenon can be attributed to multiple mechanisms, encompassing the antioxidant, anti-inflammatory, antithrombotic, and antiapoptotic properties associated with these particles [86].

HDL are endogenous particles, sizing approximately 7–12 nm, with prominent characteristics such as: (i) high diffusion capacity through the intravascular space; (ii) promotion of the reverse cholesterol transport from peripheral tissues to the liver; and (iii) cross-talking with endothelial cells by carrying several active metabolites, among which proteins, lipids, phospholipids, apoproteins and micro-RNAs (miRNAs) are in constant and intense hemodynamic traffic [87,88]. The structure of HDL consists of numerous proteins, lipids, and miRNA, organized with a hydrophobic core and a hydrophilic surface. Nevertheless, this structure undergoes continuous remodeling within the bloodstream, achieved through processes such as lipid transfer, hydrolysis, esterification, and the capture of proteins that exhibit high affinity for HDL.

Interestingly, there is evidence suggesting that HDL mimetic therapy may confer myocardial protection post-MI [89]. This effect should contribute to reducing the extent of MI in studies using ex vivo models that tested coronary reperfusion with HDL. HDL exhibits multiple cytoprotective actions that can engage various protective mechanisms against NLRP3 inflammasome activation and DAMP signaling [90]. These mechanisms can be initiated through the interaction between HDL and endothelial cells or cardiomyocytes. In essence, cardiomyocyte protection can be achieved directly via lipoprotein-to-cardiomyocyte contact, on indirectly via endothelium–cardiomyocyte contact or cross-talking [91].

While direct evidence of the interaction between HDL and the myocardium remains limited, the presence of scavenger receptor (SR-BI) expression in cardiomyocytes emerges as a significant factor contributing to the cardioprotective effect [92]. In general, the binding, internalization, and transport of HDL across endothelial cells involve three key proteins: (i) SR-BI, (ii) endothelial lipase, and (iii) the ATP-binding cassette (ABCG1). Additionally, lipid-free apoA-I undergoes lipidation by the ATP-binding cassette A1 (ABCA1) before being transported as part of the HDL complex [93]. Directly or indirectly, HDL’s cardioprotective effect, achieved by dampening inflammasome activation in the myocardium, may be attributed to various mechanisms.

## 7. The Protective Mechanisms of HDL on Cardiomyocyte Pyroptosis

HDL plays a crucial role in preserving mitochondrial function by activating intracellular protein kinase signaling through various receptors. In addition, HDL can also engage with intracellular G protein-coupled receptors (GPCRs), both trimeric and small [94]. Upon binding to SR-BI, ABCA1, or GPCRs, these receptors initiate downstream signaling cascades, influencing cellular processes like proliferation, metabolic regulation, and cell survival. In the context of reperfusion following MI, these activated pathways are collectively referred to as the reperfusion salvage kinase (RISK) [95]. The RISK pathway comprises two signaling routes mediated by the phosphoinositide 3-kinases (PI3Ks) family plus protein B (Akt) and mitogen-activated protein kinase 1, plus extracellular signal-regulated kinase 1/2 (MEK1-ERK1/2) [96]. Both are triggered by stimuli to transmembrane GPCRs.

PI3K-Akt inhibits the mitochondrial permeability transition pore (mPTP) by reducing glycogen synthase kinase 3 beta (GSK3β) activity, preserving mitochondrial integrity [97,98]. Additionally, it activates hexokinase II (HKII), essential for glycolysis during ischemic or hypoxic conditions [99,100,101,102]. MEK1-ERK1/2 stimulates anti-apoptotic mechanisms and desensitizes mPTP via the cyclophilin D pathway [103]. Furthermore, in non-cardiac cells, PI3K-Akt signaling has shown potential in regulating NLRP3 inflammasome and caspase-1 activity [104,105,106]. HDL can activate the RISK pathway during MI, thereby modulating mitochondrial DAMPs (mtDAMPs) in the cytoplasm and reducing NLRP3 inflammasome activation [107,108]. In addition, recent findings also suggest that Akt’s phosphorylation of NLRP3 on serine 5 inhibits its assembly and activity [109] (Figure 3). The phosphorylation of AKT at serine 570 may exert an inhibitory effect on peroxisome proliferator-activated receptor coactivator 1α (PGC-1α), a transcription factor responsible for mediating the dynamics of mitochondrial metabolism [110]. Possibly, a negative regulation of PGC-1α would result in increased expression of the NLRP3 inflammasome, mtDNA release, and oxidative stress [111]. Additionally, this downregulation could decrease the expression of tumor necrosis factor α-induced protein 3 (TNFAIP3), an intermediate in TLR signaling that inhibits the NLRP3 inflammasome pathway [112]. Hence, this specific AKT phosphorylation appears to have detrimental effects, which is intriguingly contradictory considering that the phosphoinositide 3-kinases (PI3Ks) family plus protein B (Akt) pathway (PI3K-Akt pathway) typically promotes cardio protection by facilitating a shift in glycolytic metabolism during ischemic events [113,114].

Another pathway activated during reperfusion, known as survivor activating factor enhancement (SAFE), is pertinent to myocardial protection. In SAFE, the key player is the signal transducer and activator of transcription 3 (STAT3) protein kinase. Typically activated via phosphorylation, tyrosine phosphorylated STAT3 can translocate to the nucleus to regulate gene expression or the mitochondrial membrane [115]. Like the RISK pathway, HDL particles can activate the SAFE pathway, illustrating their crosstalk in myocardial pro-survival mechanisms [116,117].

Serine phosphorylation of STAT3 (Ser-727 residue) is implicated in preserving respiratory complex I and inhibiting mPTP opening following MI [97,98,116]. While the STAT3 pathway is extensively studied for its role in myocardial IR protection, its direct correlation with NLRP3 inflammasome modulation remains unclear. Nevertheless, the activation of the SAFE pathway may indirectly contribute to reducing mtDAMPs by preventing NLRP3 inflammasome activation [118]. Experimental evidence with other cell types supports the feasibility of STAT3-mediated NLRP3 inflammasome signaling [119,120,121].

The SAFE pathway involves two signaling routes driven by TNF-alpha and Janus kinase 2 (JAK2)-STAT3, eventually converging downstream in the cytoplasm. TNF-alpha activates via its release and binding with the Tumor Necrosis Factor 2 Receptor (TRAF2). HDL’s interaction with TNF-alpha signaling during reperfusion, while less explored, influences TRAF2 activation, enhancing sphingosine kinase-1 (Sphk1) activity and sphingosine-1-phosphate (S1P) production, a well-studied bioactive sphingolipid, in the cytoplasm [115].

Increased S1P levels trigger intracellular TRAF2 activation, leading to further S1P production via Sphk-1 activity, and ultimately instigating JAK2-STAT3 activation through an ‘inside-out’ signaling pathway [122,123]. Inside-out signaling is believed to facilitate JAK2 activation through S1P receptors (S1PRs), which may induce phosphorylation and docking sites for STAT3 proteins, enabling their subsequent phosphorylation at serine or tyrosine residues [115].

HDL particles can activate two additional kinase pathways that require further examination in the context of MI. First, HDL may target the positive regulatory phosphorylation of NLRP3 at serine 198 in humans and serine 194 in mice by c-Jun terminal kinase (JNK) [124]. HDL has demonstrated its potential to reduce this signaling pathway’s activity, as seen in human pancreatic beta cells and endothelial cells [125,126]. Second, the protein kinase A (PKA) signaling pathway, which negatively regulates NLRP3 assembly through serine 295 phosphorylation in humans and serine 291 phosphorylation in mice [127], is associated primarily with HDL/ApoA-I via ABCA-1 and SR-BI receptors [128].

HDL particles transport sphingolipids that regulate Ca^2+^ dynamics through various mechanisms. Among these, S1P can modulate intracellular Ca^2+^ levels via S1P1–5 receptors coupled to G proteins [129]. Specifically, S1P2 and S1P3 receptors are thought to activate phospholipase C and inhibit adenylyl cyclase, facilitating intracellular Ca^2+^ mobilization [129]. Ke et al. [130] used confocal microscopy to demonstrate that S1P, in cardiomyocytes subjected to hypoxia and reoxygenation, mitigated mitochondrial Ca^2+^ overload. This effect preserved mitochondrial membrane potential by preventing the opening of the mitochondrial permeability transition pore (mPTP).

While further investigations are required to confirm these effects in the context of MI, promising results have emerged from ApoA-I infusion therapies [131]. These efforts date back to the 1980s when reconstituted HDL (rHDL), with ApoA-I as the primary protein component, was utilized [132,133]. More recently, the anti-inflammatory effects of rHDL were documented in preclinical [134,135] and translational (human ex vivo and in vivo) models, with challenges with LPS in healthy human volunteers [136]. Among these studies, the noteworthy impact of CSL-112, a recombinant HDL (rHDL), in reducing IL-1β secretion, a robust indicator of NLRP3 inflammasome pathway activation, deserves attention [137]. A comprehensive review of these studies was conducted by Morin et al. [138].

The use of rHDL therapy for patients in the acute phase of MI was evaluated in a randomized clinical trial, which showed a significant reduction in coronary atheroma volume [139]. Currently, an ongoing Phase 3 multicenter double-blind randomized placebo-controlled trial, titled ‘Study to Investigate CSL-112 in Subjects with Acute Coronary Syndrome’ (AEGIS-II; NCT03473223), holds the promise of offering further evidence regarding the therapeutic efficacy of rHDL in enhancing post-MI outcomes. While this trial was not specifically designed to assess the extent of MI, the data related to cardiovascular mortality and the incidence of heart failure could indicate the potential relevance of this mechanism.

ApoM is found not only in HDL particles but also in other lipoproteins like LDL, very low-density lipoprotein (VLDL), and chylomicrons. Its protective effect on mitochondrial function is linked to its ability to bind with S1P, forming the ApoM–S1P complex, which facilitates intravascular transport. It is noteworthy that roughly 65% of intravascular S1P is carried by HDL particles through their interaction with ApoM [140].

S1P is naturally produced intracellularly through the catalytic activity of sphingosine kinase isoforms 1 and 2 (SphK1 and SphK2), regulated by phosphorylation reactions [141]. S1P can indeed initiate signaling cascades in vascular endothelium and cardiomyocytes, offering various protective functions. Notably, this sphingolipid plays a role in a range of biological processes, encompassing cell motility, proliferation, differentiation, cytoskeleton organization, cell growth, and survival. It achieves this through paracrine and autocrine cell communication, particularly during acute pathological conditions and responses to metabolic stress, such as MI [142,143,144,145].

Exploring S1P metabolism during MI poses challenges. The transcription rate of cardiac SphK1, responsible for S1P synthesis, increases during MI [146]. Correspondingly, plasma S1P levels remain stable during the initial 2 h following MI onset, rise over the subsequent 12 h, and then revert to levels comparable to those seen in patients with stable coronary artery disease (sCAD) [147]. However, HDL-bound plasma S1P levels are lower in both MI and sCAD patients compared to controls, indicating a reduced capacity to retain S1P within the HDL of these patients [147]. The observed decline in apolipoprotein M production during the acute inflammatory response could potentially explain this discovery [148]. Nevertheless, further research is necessary to establish a definitive understanding of the interplay between MI, HDL-bound S1P and apolipoprotein M.

Preclinical studies have showcased myocardial protection against IRI by augmenting S1P levels through diverse strategies, including the inhibition of the endogenous catabolic enzyme S1P lyase [142], S1P supplementation using native or recombinant HDL particles [149], and elevating S1P availability via genetic modifications [150]. The crucial role played by S1P’s interaction with myocardial cells via S1PR1–5, G protein-coupled receptors belonging to the class A family [151], underpins these cytoprotective mechanisms. While the precise effects of S1PR1–5 are not fully elucidated, the interconnections represented by complexes like S1P–S1PR1, S1P–S1PR2, and S1P–S1PR3 are associated with shielding against acute myocardial IRI [143]. An illustrative example involves the interplay between the inhibitory regulator of G proteins and β-arrestin within the S1P1 signaling pathway. HDL particles binding to S1P exert a regulatory influence, leading to anti-inflammatory effects by inhibiting the NF-κB pathway via ERK-1/2. This ultimately holds the potential to reduce the synthesis of NLRP3 inflammasome proteins [103,152].

The connection between the apoM-S1P complex and the NLRP3 inflammasome is still a subject of ongoing research. Weigert et al. [153] shed light on another avenue of interaction between S1P metabolism and NLRP3 inflammasome activity in an in vitro study. They demonstrated that caspase-1’s proteolytic activity hinders SphK2 function by cleaving its N-terminal portion, thereby disrupting S1P metabolism. However, the in–out signaling whose interactions with the NLRP3 inflammasome are best demonstrated are due to the synthesis of S1P via SphK1. So far, it remains uncertain whether SphK1 might also play a role in this interaction.

ApoJ, also known as clusterin, is a chaperone composed of heterodimeric glycoproteins. It is primarily found in smaller, denser HDL particles, and plays a vital role in cholesterol transport and the prevention of oxidative stress caused by Reactive Oxygen Species (ROS) [154]. In a porcine model of myocardial IRI induced by the percutaneous occlusion of the left anterior coronary artery, there was an increase in the plasma concentration of ApoJ, suggesting a potential connection between ApoJ and acute cardiac injury [155]. The precise mechanism underlying this effect remains unclear. However, ApoJ exhibits antioxidant properties and can activate the RISK pathway, which implies it might be a compensatory metabolic response during the acute post-injury phase [156]. Furthermore, ApoJ’s cardioprotective effects are associated with the downregulation of the expression of tumor necrosis factor-α (TNF-α) and Bcl-2-associated X protein (BAX) through the inhibition of inflammatory signaling via the NF-κB pathway [157]. This inhibition is equally relevant for reducing the activation of the NLRP3 inflammasome and protein transcription mechanisms.

Thacker et al. [90] explored an alternative mechanism in which HDL might influence NLRP3 inflammasome activation. They conducted in vitro experiments using DAMPs mimetics. In this study, they treated human monocytes and macrophages (THP-1) with rHDL, consisting solely of ApoA-I and soybean phospholipids. The results revealed a noteworthy reduction in IL-1β levels by rHDL when these cells were exposed to cholesterol crystals, which act as danger signals capable of triggering a proinflammatory response. This suggests that HDL possesses the potential to exert anti-inflammatory effects by antagonizing NLRP3 and IL-1β transcription, as well as inhibiting caspase-1 activation. Both HDL actions on NLRP3 signaling and caspase-1 activation may prevent the disruption of lysosomal membrane integrity [90]. Typically, after coronary reperfusion, there is lysosomal degradation coupled with an accumulation of autophagosomes, stemming from disruptions in the autophagic process. These changes are intricately linked to the activation of the NLRP3 inflammasome and the subsequent demise of cardiomyocytes [158,159,160].

## 8. The Molecular Changes and Pathogenic Effects Produced by HDL during MI

As mentioned above, in the acute phase of MI, significant alterations occur in the activity of cells within the cardiovascular and immuno-inflammatory systems, leading to a profound transformation in the molecular composition of HDL [103] (Figure 4). HDL particles in patients with MI exhibit several dysfunctions when compared to individuals with stable coronary artery disease.

This transformation encompasses a reduction in particle size associated with increased density, shifts in miRNA profiles, the oxidative modification of lipids and proteins due to oxidative stress, the incorporation of inflammatory proteins, and the depletion of protective proteins. Importantly, this metamorphosis in HDL commences even prior to the occurrence of MI, prompted by cardiovascular risk factors, and intensifies in the days following the onset of MI symptoms [161]. These changes in the lipid–protein–miRNA triad transform the phenotype of HDL particles, rendering them dysfunctional and potentially pathogenic [103,162,163].

## 9. Oxidation in HDL Components during MI and Inflammasome Activation

In the early hours after MI, our research has revealed a heightened oxidation state and a decreased antioxidant capacity of HDL [164,165]. This alteration in HDL functionality is primarily attributed to its increased exposure to reactive oxygen and nitrogen species secreted by immune system cells, as well as its capacity to absorb oxidized molecules from other lipoproteins, notably LDL [166]. This intricate process can modify HDL functionality through quantitative and qualitative changes within its components, including various apolipoproteins (ApoA-I, ApoA-II, ApoA-IV, ApoD, ApoE, ApoF, ApoJ, ApoL1, and ApoM) and enzyme ligands (serum amyloid A [SAA], Paraoxonase-1 [PON1], Paraoxonase 3 [PON3], Lecithin cholesterol acyl transferase [LCAT], platelet-activating factor acetylhydrolase [PAF-AH], Glutathione peroxidase 3 [GPX3], Cholesteryl ester transfer protein [CETP], and phospholipid transfer protein [PLTP]) [167,168]. During this period of acute phase MI, when oxidative processes are heightened, there may also be shifts in the distribution of these components across different HDL subfractions. Notably, a study involving 69 patients with ST-segment elevation myocardial infarction (STEMI), in comparison to 67 healthy controls, revealed that the reduction in antioxidant capacity has a more significant impact on smaller HDL subtypes, particularly HDL3 [169]. This change was correlated with the decreased concentration of PON1 in this specific HDL subtype, a crucial enzyme in hydrolyzing oxidized LDL and mitigating the harmful effects of products resulting from phospholipid peroxidation [170]. Consequently, oxidative stress during the acute phase of MI is closely associated with the development of inflammation [171].

However, while research in atherosclerotic disease has indicated that the oxidation of methionine residues in human apoA-I can serve as a potent inflammasome activator [172], no clinical studies have definitively established a direct link between oxidized HDL components and the activation of the NLRP3 inflammasome signaling pathway during MI. Therefore, it is imperative to conduct further investigations to explore the specific pro-inflammatory mechanisms arising from oxidized HDL components.

## 10. HDL Lipid and Protein Composition Changes during MI and Inflammasome Activation

Observational studies have reported notable alterations in plasma lipid species during MI [173,174]. Specifically, in HDL, enzymes like CETP, which facilitate lipid transport among lipoproteins, exhibit heightened activity during the acute MI phase [164]. Consequently, the lipid composition of HDL undergoes significant changes, including a reduction in phospholipids, ester cholesterol, and S1P, along with an elevation in free cholesterol, triglycerides, saturated fatty acids, lysophosphatidylcholine, and lipid peroxidation products [175]. These shifts in HDL composition appear to be particularly relevant within the phospholipid category when comparing patients diagnosed with acute coronary syndrome compared to those with stable coronary disease [176]. Within the realm of phospholipids, a specific subclass known as plasmalogens emerges as particularly noteworthy [176,177]. Although the precise function of plasmalogen remains incompletely understood, its concentration, inversely related to inflammatory diseases, warrants further investigation to establish its clinical significance [177]. In pre-clinical in vivo models, dietary modifications aimed at increasing plasmalogen levels demonstrated a reduction in the inflammatory vascular marker vascular cell adhesion molecule 1 (VCAM1) [178]. Additionally, it is important to consider plasmalogen’s role as a carrier, as indicated by the significant attenuation of NLRP3 inflammasome pathways in vitro when immune microglial cells (BV-2) were pre-treated with eicosapentaenoic acid (EPA)-enriched ethanolamine plasmalogen [179].

During MI, there is a notable divergence among patients in terms of triglyceride levels, with some experiencing an increase while others experience a reduction [180]. Remarkably, the APOC-III protein levels within both LDL and HDL2 exhibit an elevation of approximately 45%, corresponding proportionately with the rise in triglycerides [181]. While the precise mechanism behind this shift in APOC-III remains elusive, its clinical ramifications are of considerable importance. Apo-CIII has been identified as a trigger for pro-inflammatory signaling, particularly via an alternate NLRP3 inflammasome pathway, as observed in human monocytes. Apo-CIII is predominantly found in very-low-density lipoproteins (VLDL) and is least abundant in HDL. However, the pro-inflammatory effects of Apo-CIII are solely elicited in the presence of VLDL alone. This is because the existence of other anti-inflammatory apolipoproteins, such as ApoA-I present in HDL, may potentially counterbalance its deleterious impact, as suggested by Zewinger et al. [182]. Furthermore, it is noteworthy that delipidated Apo-CIII has the capacity to induce the release of IL-1β from human monocytes without requiring priming with lipopolysaccharide. This induction occurs through the initiation of Toll-like receptor (TLR) 2 and TLR4 dimerization, Ca^2+^-dependent superoxide production, and the activation of caspase 8, presenting an alternative pathway for NLRP3 inflammasome activation [182].

The SAA, a prominent acute-phase inflammatory protein, is synthesized in abundance during the acute phase of MI and is primarily transported by HDL, accounting for approximately 95% of its transport [183]. The distinct cone-shaped structure of SAA, characterized by hydrophobic and hydrophilic regions on helices 1 and 3, promotes intricate molecular interactions on the lipid surface of HDL [184]. Initial investigations proposed that SAA diminishes the anti-inflammatory and antioxidative attributes of HDL, thereby compromising its functionality. However, it has since been revealed that the enrichment of HDL with SAA serves as a safeguard against lipoprotein oxidation. Furthermore, the mild oxidation of SAA-enriched HDL triggers the release of SAA, which in turn exhibits notable antioxidant properties (see this review for more details [181]). This is consistently supported by findings demonstrating that HDL derived from individuals with elevated SAA levels displays heightened antioxidant activity in comparison to control groups [185]. In summary, while an array of in vitro assays has extensively documented the impairment of HDL function during inflammation, the role of SAA in mediating HDL dysfunction in vivo lacks robust substantiation.

Delipidated human SAA can replace apoA-I within HDL, a phenomenon partially facilitated by CETP [186]. This process not only reduces circulating SAA levels but indirectly limits its bioavailability, thus mitigating its impact on detrimental pathways [187]. Furthermore, recent evidence has revealed that the incorporation of SAA into HDL prevents SAA from inducing the release of IL-1β, activating NLRP3 inflammasomes, and promoting ROS generation, potentially leading to a reduction in SAA’s proinflammatory influence [188]. Consequently, the capture of SAA can be viewed as a protective mechanism undertaken by HDL or a dampening of SAA’s pro-inflammatory activities in the circulation.

The acquisition of SAA by HDL implies the loss of ApoA-I, which can lead to a decline in the functionality of this particle. ApoA-I represents the major apolipoprotein in HDL and is essential for their formation and protective activities [189]. Research conducted in animal models has yielded compelling evidence regarding the positive impact of ApoA-I administration immediately following myocardial IRI. The introduction of ApoA-I during reperfusion sets in motion a cascade of highly beneficial processes within the myocardium. Firstly, it promotes enhanced glucose uptake by the myocardial tissue, effectively bolstering metabolic support [190]. Simultaneously, ApoA-I functions as a safeguard, shielding against post-ischemic mitochondrial damage, thereby preserving vital cellular energy production [191]. Furthermore, ApoA-I facilitates heightened perfusion in the injured myocardium by triggering the release of nitric oxide, a potent vasodilator that amplifies blood flow [192]. Intriguingly, in a mouse model of MI, ApoA-I has demonstrated a predilection for binding to pro-inflammatory monocyte subtypes. This binding, in turn, was correlated with an augmentation in anti-inflammatory subtypes, resulting in a marked enhancement in cardiac function [192]. Collectively, this body of evidence highlights the considerable potential of ApoA-I in mitigating cardiac inflammation and, consequently, its potential to attenuate the development of heart failure post-MI.

It is worth noting that not only a decrease in the availability of ApoA-I but also its displacement from HDL can yield adverse outcomes following MI. Remarkably, the ability of ApoA-I to inhibit inflammasome activation hinges on its association with HDL; lipid-free ApoA-I fails to suppress inflammasome activation. ApoA-I’s tertiary structure is known for its dynamic nature, undergoing substantial alterations contingent upon lipid content and HDL particle size. Different HDL subtypes may possess varying capacities to impede inflammasome activation due to the conformational characteristics of ApoA-I. This is particularly relevant given that HDL tends to reduce in size in patients during the acute phase of MI.

## 11. HDL microRNA Composition Change during MI

Approximately 85% of the human genome contains non-coding RNAs, with microRNAs (miRNAs) being a prominent subset consisting of roughly 22 nucleotides. MiRNAs play a pivotal role in post-transcriptional gene regulation, influencing critical processes like inflammatory responses, cholesterol homeostasis, oxidative stress, and hypertension—all of which are fundamental to cardiovascular disease [193]. Cells utilize three well-established mechanisms for miRNA secretion: (a) exosomes, (b) miRNA complexes containing Aug2, and (c) miRNA complexes associated with HDL. Notably, miRNA: HDL complexes are readily taken up by various cell types, including macrophages and cardiomyocytes [194].

During MI, changes in miRNA bound to HDL have been documented because of the systemic inflammatory response and as a direct effect of HDL. For instance, in a cohort comprising 93 individuals, HDL’s capacity to suppress pro-inflammatory miRNA expression was significantly compromised during MI [195]. HDL-mediated Stat3 activation plays a pivotal role in downregulating the expression of miR-34B and miR-337 [21]. These two specific miRNAs, miR-34B, and miR-337, exhibit heightened levels in myocardial tissues in animal models of IRI [196], as well as in the blood of patients with MI [197] or atherosclerotic coronary disease [198].

Pre-clinical and clinical investigations have provided robust validation for specific miRNAs that play pivotal roles in regulating the pro-inflammatory NLRP3 inflammasome pathway during myocardial injury, exacerbating pyroptosis and cardiac damage. These miRNAs include miR-1, miR-15, miR-29a, miR-29b, miR-30d, miR-125-5p, miR-149, miR-155, miR-383, and miR-424, as elaborated in references [199,200]. Conversely, there is an increased expression of other miRNAs associated with myocardial protection, acting as negative regulators of the NLRP3 inflammasome pathway during myocardial injury. These protective miRNAs encompass miR-9, miR-30c-5p, miR-100-5p, miR-133a-3p, miR-135b, miR-148a, miR-181b-5p, miR-223-3p, MicroRNA-330-5p, miR-320, miR-351, miR-495-3p, miR-590-3p, and miR-703, as documented in references [199,201,202,203]. Among these miRNAs, miR-21’s role remains somewhat enigmatic. While it exerts a protective effect against myocardial injury and pro-inflammatory modulation in the context of myocardial infarction, miR-21 is also implicated in the promotion of cardiac fibrosis post-MI, contributing to adverse cardiac remodeling [204,205,206]. Importantly, an intriguing hypothesis for future studies revolves around whether HDL particles have the capacity to modulate miRNA expression pathways or serve as carriers for specific miRNAs targeting cells during MI [207]. This research field holds significant promise in further elucidating the intricate mechanisms underlying cardiac injury and protection [207].

## 12. Conclusions

One of the significant repercussions of myocardial ischemia and reperfusion injury post-MI is the pronounced inability to suppress the proinflammatory cell-death pathway governed by the NLRP3 inflammasome. In essence, this damage results in the formation of pores on the plasma membrane surface, leading to the release of cytosolic components into the extracellular environment, thereby triggering the activation of the innate immune system through DAMPs motifs. HDL particles have accumulated a body of evidence demonstrating their capacity to promote myocardial protection, primarily through their anti-inflammatory properties, which involve the modulation of NLRP3 inflammasome signaling. Although several of these protective functions may be compromised during MI due to modifications in HDL structure, resulting in the formation of dysfunctional particles, the use of functional native or recombinant HDL for MI treatment is an intriguing avenue of research. It holds the potential to provide comprehensive insights into the true role of HDL in the context of MI.

## Figures and Tables

**Figure 1 ijms-25-01290-f001:**
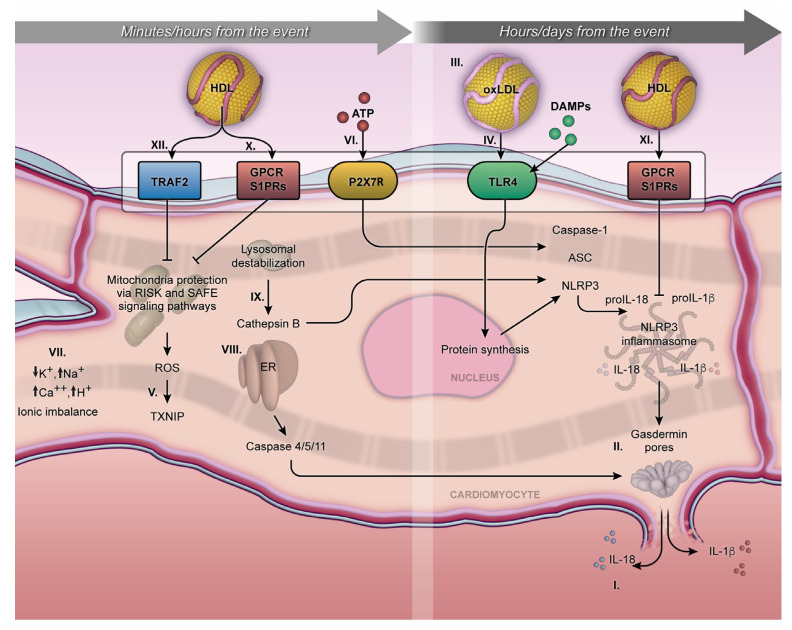
Minutes/hours from MI and hours/days from MI. The NLRP3 inflammasome cellular mechanism initiates during the first minutes of MI and can persist for days, exerting an influence on acute or chronic outcomes. Activation of pro-IL-1β and pro-IL18 and their respective release outside the membrane (I); gasdermin-D on the lytic cell death due to the pore formation on the cellular membrane, characterized by proinflammatory cell death or pyroptosis (II); proinflammatory oxidized LDL (III); TLR4 and NLRP3 inflammasome transcription pathway (IV); MI exacerbates the production of mitochondrial ROS in a process mediated by the TXNIP (V); P2X7R receptor under ATP stimulation promotes calcium and sodium influx, resulting in potassium efflux and classic signaling for NLRP3 inflammasome activation (VI); ionic imbalance (mainly by increase of Ca^2+^ mobilization and decrease of K^+^ efflux) (VII); endoplasmic reticulum stress, triggering caspase 4/5/11 activation (VIII); Cathepsin B, inducing NLRP3 activation (VIX); S1PRs play a crucial role in the interaction of S1P, facilitating cytoprotective mechanisms both short- and long-term, following MI (X and XI). The activation of the Tumor Necrosis Factor 2 Receptor (TRAF2) by HDL can influence the activity of sphingosine kinase-1 and consequently increase the production of sphingosine-1-phosphate (XII). The arrows mean the increase (↑) or decrease (↓) in ion concentration into the cytoplasmic milieu.

**Figure 2 ijms-25-01290-f002:**
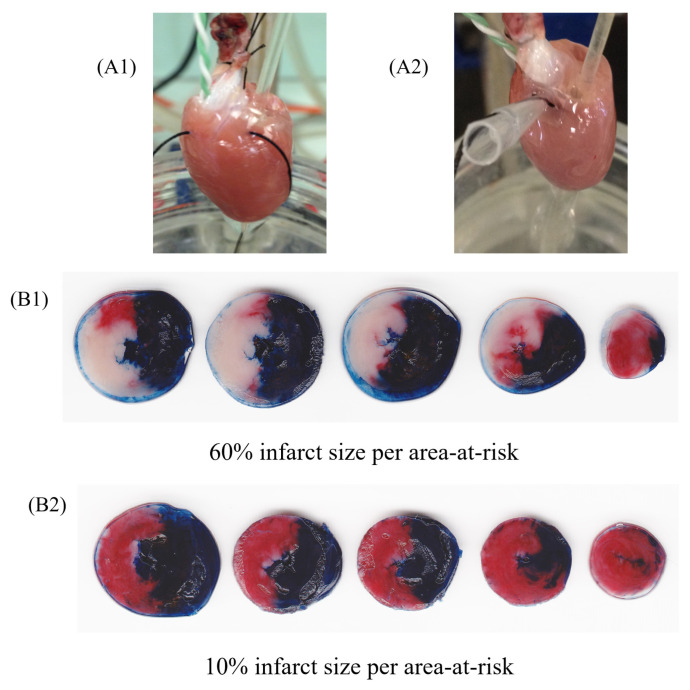
Representative image of isolated perfused hearts (ex vivo) using Langendorff apparatus undergoing ischemia and reperfusion protocol. Rat heart during regional ischemia (**A1**) and reperfusion (**A2**). Figures represent groups that received vehicle (**B1**) or HDL (200 µg/mL) infusion (**B2**), after the protocol of staining with 2,3,5-triphenyltetrazolium chloride staining. The heart sections (**B1**,**B2**) are presented in tree colors for histologic analysis: blue, the non-ischemic area (also known as non-risk area); red, the ischemic area (also known as area-at-risk); and white or pale, the infarct area. The infarct size was calculated as a percentage of the area-at-risk.

**Figure 3 ijms-25-01290-f003:**
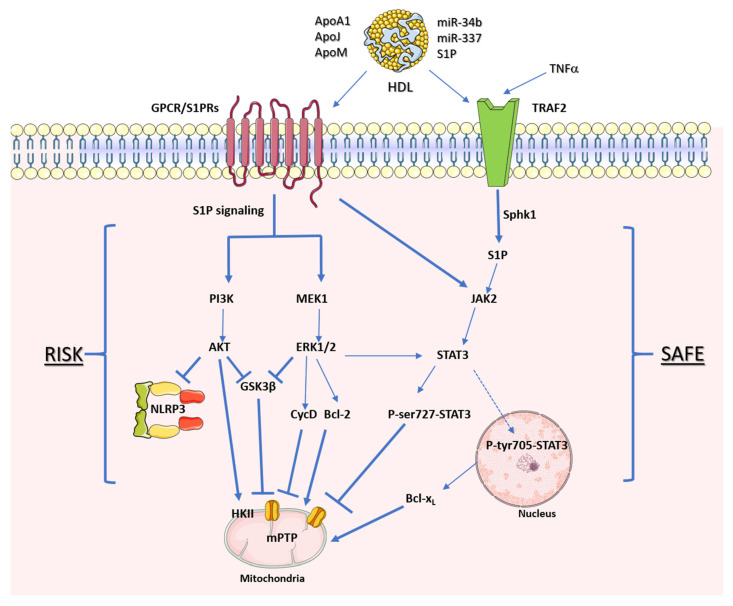
HDL receptors of cardioprotective protein kinase pathways. Representative signaling mediated by HDL through the reperfusion salvage kinase (RISK) pathway and the survivor-activating factor enhancement (SAFE) pathway. The arrows represent stimulation, while the hammerheads represent inhibition.

**Figure 4 ijms-25-01290-f004:**
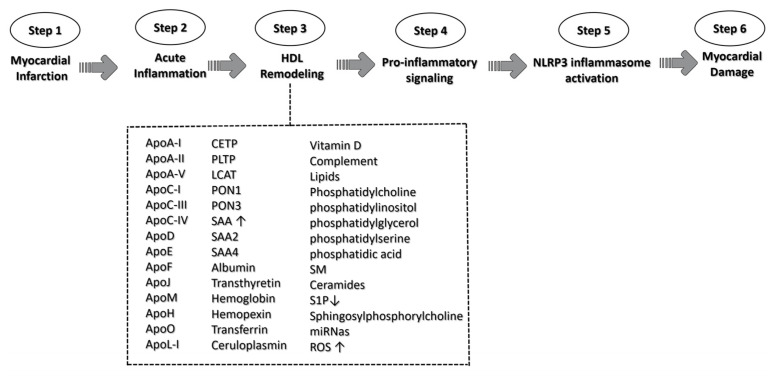
A schematic representation of HDL triggering NLRP3 activation signaling. The role of HDL in the context of myocardial infarction is illustrated along the six pivotal steps, culminating in NLRP3 activation signaling and the ultimate myocardial damage. HDL complement acronyms that could participate in the remodeling process: apolipoprotein A-I (ApoA-I); apolipoprotein A-II (ApoA-II); apolipoprotein A-V (ApoA-V); apolipoprotein C-I (ApoC-I); apolipoprotein C-III (ApoC-III); apolipoprotein C-IV (ApoC-IV); apolipoprotein D (ApoD); apolipoprotein E (ApoE); apolipoprotein F (ApoF); apolipoprotein J (ApoJ); apolipoprotein M (ApoM); apolipoprotein H (ApoH); apolipoprotein O (ApoO); apolipoprotein (ApoL-I); cholesteryl ester transfer protein (CETP); phospholipid transfer protein (PLTP); lecithin cholesterol acyltransferase (LCAT); paraoxonase-1 (PON1); paraoxonase-1 (PON3); serum amyloid A (SAA); serum amyloid A-2 (SAA2); serum amyloid A-4 (SAA4); Sphingomyelin (SM); Sphingosine 1-phosphate (S1P); microRNA (miRNas); and reactive oxygen species (ROS). ↑ means increase; ↓ means decrease.

## Data Availability

Not applicable.

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
