# Peer review of "High-Density Lipoproteins at the Interface between the NLRP3 Inflammasome and Myocardial Infarction"

_ijms, 2024, doi:10.3390/ijms25021290_

Round 1
Reviewer 1 Report
Comments and Suggestions for Authors
Dear authors: Congratulations for you excellent review article. I have only minor suggestions:
-In lines 36-37 you wrote: as sustantiated by numerous studies [4-7]. I suggest changing several instead of numerous. This is because you refered to only 4 studies.
- In the Introduction, the term " reperfusion therapies" is used several times. I recommend making a very brief description of the meaning of this term for a better understanding.
- The authors have presented an excellent description of the immunological mechanisms activated during myocardial infarction, and also of the protective effects of HDL on various immunological signaling pathways. This detailed description will surely improve our understanding of the complex pathophysiology of myocardial infarction.
- In lines 542-546 you wrote: " In the early hours after MI, our research has revealed a heightened oxidation state and a decreased antioxidant capacity of HDL [158,159]. This alteration in HDL functionality is primarily attributed to its increased exposure to reactive oxygen and nitrogen species secreted by immune cells....
Question: In addition to the reperfusion technique, do you think it would be convenient to add some pharmacological intervention that would inhibit the excessive production of reactive oxygen species? For example, nitric oxide synthase inhibitors?
Author Response
Reviewer #1
Dear authors: Congratulations for you excellent review article. I have only minor suggestions:
* - In lines 36-37 you wrote: as sustantiated by numerous studies [4-7]. I suggest changing several instead of numerous. This is because you refered to only 4 studies.
Response from authors: We appreciated the careful reading and the comments made. The change from numerous to several was made so that we are more consistent with the number of articles cited. From line 34 to 37, as follows: “The extent of MI is a potent predictor of various adverse cardiovascular events, encompassing mortality, recurrent MI, arrhythmias, congestive heart failure, angina, and the need for revascularization, as substantiated by several studies [4-7]”.
*- In the Introduction, the term " reperfusion therapies" is used several times. I recommend making a very brief description of the meaning of this term for a better understanding.
Response from authors: We thank the reviewer for the helpful suggestion. We included a brief description and examples of “reperfusion therapies” to enhance the reader’s understanding. From line 38 to 44, as follows: “In other words, reperfusion therapies aim to restore blood flow by unclogging an occlusive thrombus, mostly resulting from the rupture of an atherosclerotic plaque. This is achieved through two primary methods: non-invasive, which involve the use of thrombolytic drugs, and invasive procedures such as primary percutaneous coronary intervention (PCI) or coronary artery bypass graft (CABG) surgery. These methods allow blood reperfusion and aid in the restoration of appropriate circulation.”
*- The authors have presented an excellent description of the immunological mechanisms activated during myocardial infarction, and also of the protective effects of HDL on various immunological signaling pathways. This detailed description will surely improve our understanding of the complex pathophysiology of myocardial infarction.
Response from authors: We appreciate the reviewer's kindly reading and valuable comments on our review. It certainly encourages us to continue our studies on the pathophysiology of myocardial infarction.
*- In lines 542-546 you wrote: " In the early hours after MI, our research has revealed a heightened oxidation state and a decreased antioxidant capacity of HDL [158,159]. This alteration in HDL functionality is primarily attributed to its increased exposure to reactive oxygen and nitrogen species secreted by immune cells....
* Question: In addition to the reperfusion technique, do you think it would be convenient to add some pharmacological intervention that would inhibit the excessive production of reactive oxygen species? For example, nitric oxide synthase inhibitors?
Response from authors: We appreciate the reviewer's relevant question. The pharmacological intervention, also known as pharmacologic reperfusion, have shown promising results in preclinical studies. However, translating these findings into clinical practice has proven more challenging. This is particularly evident in cases where pre-existing conditions like advanced atherosclerosis or diseases with an inflammatory background (e.g., diabetes mellitus). To overcome these challenges and effectively translate the benefits from preclinical studies, it is crucial to improve experimental models to better represent the profile of the patient we wish to help.

Reviewer 2 Report
Comments and Suggestions for Authors
The review entitled “HDL at the interface between the NLRP3 inflammasome and myocardial infarction” describes extensively the role of inflammasome activation in the context of myocardial infarction (MI) and provide a complete analysis of how HDL might modulate this process. The review is really impressive and updated. However, the conclusion is a little bit frustrating for readers, as it is just concluded that the use of functional native or recombinant HDL for MI treatment is "an intriguing avenue of research". After so long review, I expected something else. In turn, the goal of the review is really starting at section 6, so that the first sections could be reduced.
The following minor points would be addressed before definitive acceptance.
Line 57: A brief explanation about AKA alarmins would be convenient.
Line 75 on: Some mechanisms included at the figure are not described at the legend, so that some proteins are not discussed (TRAF2, P2X7...). Perhaps the presentation of Figure 1 is premature, as these proteins are discussed in posterior sections. As DL S1PRs is involved at short and long periods after MI, and the links between HDL and MI is the main topic of the review, this point would be highlined.
Line 122: At that point, some explanation about the generation of oxLDL, the plasma level of paraoxonases and the mechanims for enhancing the inflammatory reponse is needed.
Line 170-on: The electrical charge of ions ( Ca2+, Na+, K+) would be written as superscript.
Line 528: Figure 4 should be improved, specially step 3 about FDL remodeling. The increased and decreased components and the mechanism of action should be detailed.
Author Response
Reviewer #2
The review entitled “HDL at the interface between the NLRP3 inflammasome and myocardial infarction” describes extensively the role of inflammasome activation in the context of myocardial infarction (MI) and provide a complete analysis of how HDL might modulate this process. The review is really impressive and updated. However, the conclusion is a little bit frustrating for readers, as it is just concluded that the use of functional native or recombinant HDL for MI treatment is "an intriguing avenue of research". After so long review, I expected something else. In turn, the goal of the review is really starting at section 6, so that the first sections could be reduced.
The following minor points would be addressed before definitive acceptance.
*Line 57: A brief explanation about AKA alarmins would be convenient.
Response from authors: We thank the reviewer’s consideration and for the meticulous reading of our study. We changed the abbreviation from AKA to a.k.a. (also known as), to make it clearer to the readers that both words (DAMPs and alarmins) are equivalent. From line 61 to 63, as follows: “The NLRP3 inflammasome is an intracellular defender that acts against cellular fragments and cytoplasmic contents of injured and/or dead cells, called damage-associated molecular patterns (DAMPs) (a.k.a alarmins)”.
*Line 75 on: Some mechanisms included at the figure are not described at the legend, so that some proteins are not discussed (TRAF2, P2X7...). Perhaps the presentation of Figure 1 is premature, as these proteins are discussed in posterior sections. As DL S1PRs is involved at short and long periods after MI, and the links between HDL and MI is the main topic of the review, this point would be highlined.
Response from authors: We thank the reviewer for the comments and the sensible suggestions. We included the description of the TRAF2 receptor and the legend of Figure 1, identified with the Roman numeral XII, as suggested. From line 100 to 102, as follows: “The activation of the Tumor Necrosis Factor 2 Receptor (TRAF2) by HDL can influence the activity of sphingosine kinase-1 and consequently increase the production of sphingosine-1-phosphate (XII).”
Also, as pointed out, we corrected the transcription of P2X7 to P2X7R. From lines 95 to 97, as follows: “P2X7R receptor under ATP stimulation, promotes calcium and sodium influx, resulting in potassium efflux, classic signaling for NLRP3 inflammasome activation (VI).”
Regarding Figure 1, as it concentrates on some mechanisms discussed since the introduction, we decided to present it with some anticipation. We hope this option can improve the readers’ understanding.
Finally, we also added a description in the caption highlighting the role of S1PRs receptors in short and long periods after MI, indicated by the Roman numeral X and XI. From line 99 to 100, as follows: “S1PRs play a crucial role in the interaction of S1P, facilitating cytoprotective mechanisms both in the short- and long-term following MI (X and XI).”
Line 122: At that point, some explanation about the generation of oxLDL, the plasma level of paraoxonases and the mechanims for enhancing the inflammatory reponse is needed.
Response from authors: We appreciated the sensible suggestion. We added in the text a reference to the role of paraoxonase, especially paraxonase-1 and its effects inversely related to the inflammatory manifestation induced by ox-LDL. From line 140 to 145, as follows: “Nevertheless, the antioxidant function of the paraoxonase family, notably paraxonase-1 (PON1) within HDL, significantly mitigates the inflammatory impact mediated by ox-LDL. Animal model experiments have indicated a direct correlation between the absence of the PON1 gene and inflammatory responses. In humans, plasma levels of PON1 exhibit an inverse correlation with the development of atherosclerosis [25].”
Line 170-on: The electrical charge of ions (Ca2+, Na+, K+) would be written as superscript.
Response from authors: We thank the reviewer for the suggestion. We changed the electrical charge of ions Ca2+, Na+, and K+ as superscripts on the indicated line and throughout the manuscript. From line 192 to 206, where else necessary, as follows: “Mitochondrial Ca2+ overload has emerged as a contributor to the activation of the NLRP3 inflammasome during ischemia [42]. An excessive or sustained increase in mitochondrial Ca2+ levels can lead to mitochondrial damage and cell death [43]. During MI, damaged cardiac cells can release ATP from their membranes, initiating the activation of the P2X7R. This receptor, an ATP-gated extracellular ion channel known for its involvement in transmembrane ion migration [44, 45], responds to ATP by promoting the influx of Ca2+ and Na+ while inducing K+ efflux. This classic signaling pattern triggers the activation of the NLRP3 inflammasome [38] (Figure 1). Consequently, extracellular ATP binding to P2X7R rapidly disrupts the cytosolic ion concentration gradient.
The details of K+ efflux in NLRP3 inflammasome activation remain less comprehensively understood and appear to be independent of Ca2+ mobilization. For instance, Katsnelson et al. [46] disrupted the cytosolic concentration gradient by employing BAP-TA, a potent Ca2+ chelator and cytosolic Ca2+ buffer. Their findings demonstrated that cellular K+ efflux during NLRP3 activation operates independently of intracellular Ca2+ dynamics [44].
Line 528: Figure 4 should be improved, specially step 3 about FDL remodeling. The increased and decreased components and the mechanism of action should be detailed.
Response from authors: We fully agreed with the reviewer's comment and suggestion. We revised Figure 4 to cover most of the components related to HDL that could participate in the remodelling process. From line 546 to 555, as follows: “HDL complement acronyms that could participate in the remodelling process: apolipoprotein A-I (ApoA-I), apolipoprotein A-II (ApoA-II), apolipoprotein A-V (ApoA-V), apolipoprotein C-I (ApoC-I), apolipoprotein C-III (ApoC-III), apolipoprotein C-IV (ApoC-IV), apolipoprotein D (ApoD), apolipoprotein E (ApoE), apolipoprotein F (ApoF), apolipoprotein J (ApoJ), apolipoprotein M (ApoM), apolipoprotein H (ApoH), apolipoprotein O (ApoO), apolipoprotein (ApoL-I), cholesteryl ester transfer protein (CETP), phospholipid transfer protein (PLTP), lecithin cholesterol acyltransferase (LCAT), paraoxonase-1 (PON1), paraoxonase-1 (PON3), serum amyloid A (SAA), serum amyloid A-2 (SAA2), serum amyloid A-4 (SAA4), Sphingomyelin (SM), Sphingosine 1-phosphate (S1P), microRNA (miRNas), and reactive oxygen species (ROS).”

Reviewer 3 Report
Comments and Suggestions for Authors
Comments to the author:
The Manuscript " HDL at the interface between the NLRP3 inflammasome and
myocardial infarction. The manuscript suggest that the the role of inflammasome activa
tion in the context of MI and provide a detailed analysis of how HDL can modulate this process.
Revisions required:
1- Kindly elaborate the functions of accelerate the production of pro-inflammatory cytokines IL-1β and IL-18, leading to the inflammatory response these protiens in your reviews
2- PGC-1α-induced inactivation of the NLRP3 inflammasome via modulation of mitochondrial viability and dynamic please connect HDL with PGC 1 alpha
3- UCP1, irisin and FGF21 are very strong biomarkers for HDL related activity in terms of NLRP3
Comments on the Quality of English LanguageMinor changes are required
Author Response
Reviewer #3
Comments to the author:
The Manuscript " HDL at the interface between the NLRP3 inflammasome and
myocardial infarction. The manuscript suggests that the the role of inflammasome activation in the context of MI and provide a detailed analysis of how HDL can modulate this process.
Revisions required:
*1- Kindly elaborate the functions of accelerate the production of pro-inflammatory cytokines IL-1β and IL-18, leading to the inflammatory response these proteins in your reviews
Response from authors: We thank the reviewer for the great suggestion. We added to the manuscript the developments regarding accelerating the production of pro-inflammatory cytokines IL-1β and IL-18. From line 69 to 77, as follows: “The increased activity of the NLRP3 inflammasome leads to an exacerbated production of IL1β and IL18, responsible for an acute immune and inflammatory response. This effect can manifest locally as myocardial inflammation or systemically targeting vascular endothelium and multiple organs. The presence of IL1 receptors in immune cells facilitates the paracrine and autocrine effects of these proteins, potentially inducing increased transcription of proinflammatory genes, amplifying their synthesis and secretion [10]. Detrimental clinical effects such as fever, hypotension, myocardial contractility dysfunction, and increased ischemic events are mainly associated with elevated plasma IL1β levels.”
*2- PGC-1α-induced inactivation of the NLRP3 inflammasome via modulation of mitochondrial viability and dynamic please connect HDL with PGC 1 alpha
Response from authors: We thank the reviewer for this important contribution. We included an alternative outcome of AKT activation influenced by HDL in the text. We aim to meet the reviewer's expectations with this modification. “From line 400 to 410, as follows: Phosphorylation of AKT at serine 570 may exert an inhibitory effect on peroxisome proliferator-activated receptor coactivator 1α (PGC-1α), a transcription factor responsible for mediating the dynamics of mitochondrial metabolism [110]. Possibly, a negative regulation of PGC-1α would result in increased expression of the NLRP3 inflammasome, mtDNA release, and oxidative stress [111]. Additionally, this downregulation could decrease the expression of tumor necrosis factor α-induced protein 3 (TNFAIP3), an inter-mediate in TLR signaling that inhibits the NLRP3 inflammasome pathway [112]. Hence, this specific AKT phosphorylation appears to have detrimental effects, which is intriguingly contradictory considering that the PI3K-AKT pathway typically promotes cardioprotection by facilitating a shift in glycolytic metabolism during ischemic events [113, 114].”
*3-UCP1, irisin and FGF21 are very strong biomarkers for HDL related activity in terms of NLRP3
Response from authors: We appreciated the careful analysis and suggestion. Although we found studies that suggest that the absence of uncoupling protein 1 (UCP1) correlates with an overactivation of the NLRP3 inflammasome, we did not find any clear association with HDL and myocardial infarction. Therefore, we decided not to include information about UCP1 in this review, as it is beyond our scope of knowledge.
Fibroblast growth factor 21 (FGF21) may reduce oxidative stress, pyroptosis, and atherosclerosis by inhibiting the NLRP3 inflammasome-mediated vascular endothelial cells. We found a study in monkeys where administration of FGF21 resulted in an increase in high-density lipoprotein levels (10.1210/en.2006-1168). Also, the level of serum FGF21 in patients with T2DM positively correlated with markers of early vascular injury and high-density lipoprotein cholesterol (HDL-C) (10.1089/met.2017.0179). However, we did not find any clear relationship between HDL and myocardial infarction. For this reason, we have decided not to include in the review as, to the best of our knowledge, there is no sufficient evidence available."
Irisin is a peptide hormone released into circulation by cleavage of the fibronectin type III domain-containing protein 5 (FNDC5). It is linked to inflammation and inhibition of NLRP3 inflammasome signaling via ROS (10.1007/s10753-017-0685-3). Although some studies show a positive correlation with HDL-C levels, we have not found evidence that these changes are involved myocardial protection mechanisms. We have cautiously decided not to include it in the review due to the lack of clear relationship between HDL and myocardial infarction.
